# Integrating Macrophages into Human-Engineered Cardiac Tissue

**DOI:** 10.3390/cells14171393

**Published:** 2025-09-06

**Authors:** Yi Peng Zhao, Barry M. Fine

**Affiliations:** Department of Medicine, Division of Cardiology, Columbia University Irving Medical Center, New York, NY 10032, USA

**Keywords:** iPS, macrophage, cardiac, immune cell, engineered heart tissue

## Abstract

Heart disease remains a leading cause of morbidity and mortality worldwide, necessitating the development of in vivo models for therapeutic development. Advances in biomedical engineering in the past decade have led to the promising rise of human-based engineered cardiac tissues (hECTs) using novel scaffolds and pluripotent stem cell derivatives. This has led to a new frontier of human-based models for improved preclinical development. At the same time, there has been significant progress in elucidating the importance of the immune system and, in particular, macrophages, particularly during myocardial injury. This review summarizes new methods and findings for deriving macrophages from human pluripotent stem cells (hPSCs) and advances in integrating these cells into cardiac tissue. Key challenges include immune cell infiltration in 3D constructs, maintenance of tissue architecture, and modeling aged or diseased cardiac microenvironments. By integrating immune components, hECTs can serve as powerful tools to unravel the complexities of cardiac pathology and develop targeted therapeutic strategies.

## 1. Introduction

Myocardial infarction (MI), commonly known as a heart attack, occurs when blood flow to the heart muscle is blocked or severely reduced, leading to cardiomyocyte injury and electromechanical impairment [1]. In the United States, approximately 805,000 people experience an MI each year. Among them, 605,000 cases are first-time heart attacks, while 200,000 are recurrent events. On average, one MI occurs every 40 s [2]. A global study found MI prevalence to be 3.8% in individuals under 60 and 9.5% in those over 60 [3]. These numbers highlight the serious impact of MI and the need for ongoing research into heart injury and repair.

The heart consists of various cell types, each with a crucial role in function and recovery after injury. Cardiomyocytes, the muscle cells responsible for electromechanical contraction, make up approximately 30% of the myocardial cell population [4,5]. Endothelial cells, which form the lining of blood vessels, account for 53-64% of non-myocyte cells [4]. They regulate vascular tone, blood flow, and inflammatory responses. Cardiac fibroblasts, comprising less than 20% of non-myocytes, maintain structural integrity by producing and remodeling the extracellular matrix (ECM) [4]. Immune cells, including macrophages and lymphocytes, make up 5-10% of non-myocyte populations and fluctuate in response to injury [5]. Pericytes and vascular smooth muscle cells, which support the vasculature, account for 5-7% of the non-myocyte compartment [5].

Understanding immune cell involvement in cardiac repair is key to improving treatments for MI and other heart conditions. During an MI, macrophages and other immune cells coordinate inflammation and tissue repair. They clear dead cells, regulate fibrosis, and promote healing. Their interactions with fibroblasts and endothelial cells shape the extent of cardiac recovery [6,7,8]. Despite their critical role, many lab-based heart disease models lack immune components, limiting their accuracy in fully recapitulating real-life heart injuries.

Recent advancements in stem cell research have made it possible to generate immune cells, like macrophages, from human pluripotent stem cells (hPSCs) [9,10,11]. This breakthrough allows scientists to incorporate immune cells into disease models. One promising application is the development of human pluripotent stem cell (hPSC)-derived human-engineered cardiac tissues (hECTs), which aim to create more realistic heart models for research [12,13]. By integrating hPSC-derived macrophages into engineered tissues, scientists are poised to better replicate heart–immune system interactions and capture a key regulatory component of tissue repair and disease development.

## 2. Myocardial Injury and the Immune System

Myocardial infarction (MI) is defined as acute myocardial injury and the rise of abnormal cardiac biomarkers due to ischemia. When blood flow is suddenly disrupted, myocardial ischemia leads to cell death through necrosis and apoptosis [1,6,7]. This damage leads to heart failure and arrhythmias through attrition of cardiomyocytes and resulting replacement fibrosis. Following an MI, the heart undergoes three major phases of repair: the inflammatory phase, the reparative phase, and the proliferative phase (Figure 1) [7]. These phases highlight the critical role of immune cells in the development of heart disease and the response to injury. 

### 2.1. Inflammation Phase

The inflammatory phase following MI begins immediately as necrotic cardiomyocytes release damage-associated molecular patterns (DAMPs), including high mobility group box 1(HMGB1), heat shock proteins, mitochondrial DNA, and ATP [14,15,16,17,18]. These DAMPs trigger innate immune receptors, including pattern recognition receptors (PRRs) like toll-like receptors (TLRs), initiating a robust inflammatory response [19]. The complement cascade is also activated, which can amplify the recruitment of immune cells to the injury site [20].

Neutrophils are the first immune cells to infiltrate the infarcted myocardium [21]. They release reactive oxygen species (ROS), proteases, and inflammatory cytokines, which are critical for degrading necrotic tissue and clearing cellular debris [22]. Neutrophils also recruit additional immune cells through chemokine signaling, ensuring an organized inflammatory response. However, excessive neutrophil activity can exacerbate tissue damage through collateral oxidative stress.

As inflammation progresses, circulating blood monocytes are recruited to the injury site, driven by chemokines such as chemokine ligand 2 (CCL2/MCP-1) [23]. These monocytes differentiate into macrophages, which initially adopt a pro-inflammatory (M1) phenotype [24]. M1 macrophages produce cytokines like tumor necrosis factor (TNF-α) and interleukin-1 beta (IL-1β). These cytokines sustain the inflammatory milieu and further facilitate the clearance of dead cells and extracellular matrix debris [25,26]. 

The inflammatory phase is vital for preparing the infarcted myocardium for subsequent healing, but its duration and intensity must be tightly regulated. Prolonged or excessive inflammation can lead to adverse remodeling and increase the risk of heart failure [27].

### 2.2. Reparative and Proliferative Phase

With the resolution of inflammation, the reparative phase begins, marked by the suppression of pro-inflammatory signals and the activation of pathways that promote tissue repair and fibrosis. Anti-inflammatory (M2) macrophages play a pivotal role in transitioning from the inflammatory phase to the reparative phase by secreting key cytokines such as interleukin 10 (IL-10) and transforming growth factor-beta (TGF-β), suppressing inflammation [28,29]. Additionally, other cytokines secreted by the M2 macrophages stimulate fibroblast proliferation and differentiation into myofibroblasts, which are responsible for extracellular matrix (ECM) deposition [30,31,32]. These include the TGF-β, vascular endothelial growth factor (VEGF), and platelet-derived growth factor (PDGF).

Endothelial cells also play a significant role in the reparative phase by promoting neovascularization, where new vascular structures are assembled, a process critical for sustaining myocardial function [33]. VEGF and angiopoietins released by macrophages and fibroblasts stimulate endothelial cell proliferation and migration, leading to the formation of new capillaries [34,35,36]. These new vessels ensure the delivery of oxygen and nutrients to the regenerating myocardium, thereby supporting long-term tissue repair.

As the reparative phase progresses, the proliferative phase initiates, which is characterized by the maturation of fibroblasts into myofibroblasts and the continued refinement of scar tissue [7,37]. Regulatory T cells (Tregs) are also key modulators in the proliferative phase, suppressing prolonged inflammation and promoting tissue homeostasis [38]. Tregs secrete interleukin-10 (IL-10) and TGF-β, which not only reduce excessive immune activation but also enhance fibroblast function and contribute to balanced ECM turnover [39,40]. Other immune cells, such as dendritic cells, play an immunoregulatory role as well by modulating adaptive immune responses [41]. This ensures that excessive inflammation does not disrupt the reparative phase. 

The reparative and proliferative phases involve a complex interplay between immune cells, fibroblasts, and endothelial cells, all working in concert to stabilize the infarcted myocardium and prevent adverse remodeling. Among these, immune cells—particularly macrophages and Tregs—play a central role in facilitating and orchestrating the transition from inflammation to repair, making them attractive therapeutic research targets for improving post-MI healing and reducing the risk of heart failure. 

## 3. Cardiac Resident Macrophages

Cardiac resident macrophages, originating predominantly during embryonic development from yolk-sac progenitors, are essential for maintaining cardiac homeostasis and orchestrating responses to heart injury (Figure 2) [42,43,44,45]. Distinct from circulating monocyte-derived macrophages, these resident populations are critical regulators of cardiac physiology. They directly interact with cardiomyocytes via gap junction proteins such as connexin-43 (Cx43), facilitating electrical conduction and synchronous heart contractions [46]. Depleting these macrophages disrupts electrical conduction and can lead to conduction abnormalities, underscoring their importance in cardiac electrophysiology and overall function.

Beyond electrical conduction, cardiac resident macrophages actively regulate tissue repair mechanisms. They possess strong phagocytic capabilities, allowing them to efficiently clear apoptotic cardiomyocytes and cellular debris, particularly through the receptor Mer tyrosine kinase (Mertk) [47,48]. Failure in this clearance process exacerbates myocardial damage and hinders effective cardiac repair post-myocardial infarction (MI). Additionally, resident macrophages exert anti-fibrotic and pro-angiogenic effects, crucial for balanced cardiac remodeling and preserving heart function after injury [42]. Studies have demonstrated that depletion of resident macrophages leads to increased fibrosis and impaired angiogenesis, underscoring their protective role during cardiac stress. The following section will explore the latest technologies for generating hPSC-derived macrophages, specifically focusing on their relevance and potential applications in advancing cardiac research. 

## 4. Generation of hPSC-Derived Macrophages

### 4.1. General Methods to Achieve Mesoderm Commitment and Hemogenic Endothelium Specification

Human pluripotent stem cells (hPSCs), including both induced pluripotent stem cells (iPSCs) and human embryonic stem cells (hESCs), first differentiate into mesodermal progenitors. These progenitors then transition into hemogenic endothelial (HE) cells. Mesodermal progenitors serve as precursors for hematopoietic and macrophage differentiation. There are three major methods for differentiation: monolayer 2D cell culture, co-culture with stromal cells, and embryoid body (EB) formation (Table 1). Studies have compared these protocols in more detail [9,10,11,49,50]. 

#### 4.1.1. Monolayer 2D Cell Culture

In the monolayer two-dimensional (2D) culture method, iPSCs differentiate into macrophages through a stepwise protocol (Figure 3). This approach typically involves feeder-free conditions using Matrigel-coated plates [51,52,53]. The differentiation process mimics early embryonic hematopoiesis by guiding cells through mesoderm induction, hematopoietic specification, hematopoietic progenitor expansion, and macrophage maturation.

Mesodermal Induction (Approximately Day 0–2): Differentiation begins by directing iPSCs toward mesoderm, the germ layer responsible for hematopoietic cell generation. Initially, cells maintained in pluripotency medium (such as mTeSR1) are transferred into differentiation medium containing crucial signaling molecules. Bone Morphogenetic Protein 4 (BMP4) initiates mesoderm formation, and Activin A enhances mesoderm commitment [51,53,54]. The small molecule CHIR99021, a WNT pathway activator, is commonly used at this stage to improve mesoderm formation efficiency [52,53,55].

Hematopoietic Specification (Approximately Day 2–4): After mesoderm induction, cells enter the hematopoietic specification stage. Cells are cultured in hematopoietic differentiation medium with crucial growth factors. Vascular Endothelial Growth Factor (VEGF) promotes formation of hemogenic endothelium, a key intermediate stage producing hematopoietic cells [53,54,55,56]. Basic Fibroblast Growth Factor (bFGF) and Stem Cell Factor (SCF) further support hematopoietic differentiation. While VEGF, bFGF, and SCF are almost consistently included across the protocols, variations do exist. For example, in some protocols, cultured cells under standard oxygen (21% O2), while other protocols utilized low oxygen (5% O2) during early differentiation days [52,53,54,55].

Hematopoietic Progenitor Expansion (Approximately Day 4–7): During this phase, hematopoietic progenitors proliferate significantly. Cultures are supplemented with cytokines like VEGF, SCF, Interleukin-3 (IL-3), and Interleukin-6 (IL-6) [53,54,55]. This combination promotes proliferation and early macrophage lineage commitment. Cells begin expressing markers such as CD43 and CD45, indicative of early hematopoietic progenitors [53,55,56].

Macrophage Differentiation (Approximately Day 7–14): Expanded progenitor cells transition into macrophage differentiation medium. Macrophage Colony-Stimulating Factor (M-CSF) is critical at this stage and drives macrophage lineage specification [53,54,55]. Interleukin-34 (IL-34) is frequently added alongside M-CSF to promote tissue-specific macrophage characteristics and enhance differentiation [51,52,53].

Final Maturation (Approximately After Day 14): Macrophages continue to mature, acquiring fully differentiated macrophage or microglia-like phenotypes. Maturation medium typically includes continuous M-CSF and IL-34 supplementation. Mature macrophages express characteristic markers such as Integrin alpha M (CD11b), ionized calcium-binding adapter molecule 1 (IBA1), CD14, CD68, and additional macrophage-specific markers like TMEM119 and P2RY12 [51,52,53,54]. These cells closely resemble primary human macrophages, confirming their suitability for research and disease modeling.

Protocol Variations and Considerations: Although monolayer protocols share core steps, variations exist in cytokine use, oxygen conditions, and growth factor timing. CHIR99021 is widely used, but specific concentrations vary across protocols [51,53,54,55]. Additionally, oxygen levels during culture vary between standard (21% O2) and reduced oxygen (5% O2), impacting differentiation outcomes [52,53,54,55].

The monolayer 2D culture method provides significant advantages, such as precise control over differentiation conditions, reproducibility, and efficient macrophage generation suitable for high-throughput research [51,52,53,54,55]. However, it requires optimization of cytokine concentrations and culture conditions. The differentiation efficiency can vary significantly depending on the specific iPSC line used. Furthermore, full maturation and tissue-specific functionality might require additional molecular or environmental cues beyond standard protocols.

#### 4.1.2. Co-Culture with Stromal Cells

This method relies on OP9 mouse stromal cells, which are derived from mouse bone marrow. These stromal cells facilitate differentiation through direct cell-to-cell interactions and secreted niche factors, guiding hPSCs, including both iPSCs and hESCs, toward early mesodermal commitment and subsequent hematopoietic specification (Figure 4) [57,58,59]. 

Mesodermal Induction and Hematopoietic Specification (Approximately Day 0–9): hPSCs are generally seeded onto a monolayer of OP9 stromal cells in a medium such as the α-minimum essential medium (α-MEM) supplemented with fetal bovine serum (FBS). Cytokines are usually not added at this initial stage, as OP9 cells alone are sufficient to guide differentiation toward mesoderm through direct cell-to-cell interactions and paracrine signaling. This promotes mesodermal fate and initial hematopoietic differentiation. Early hematopoietic progenitor cells, expressing markers such as CD34 and CD43, typically emerge between days 6–9 [57,58,59,60,61,62].

Macrophage Differentiation (Approximately After Day 12): Hematopoietic progenitors harvested from the OP9 co-culture are further differentiated into macrophages in medium supplemented with macrophage colony-stimulating factor (M-CSF) [57,60]. Some protocols include granulocyte-macrophage colony-stimulating factor (GM-CSF) [60,62]. M-CSF specifically directs progenitor cells toward macrophage lineage commitment and supports their differentiation into monocytes/macrophages. GM-CSF enhances macrophage maturation and functional capabilities, such as phagocytosis. During this stage, developing cells progressively exhibit macrophage morphology and characteristics. Additionally, the combination of interleukin-3 (IL-3) and stem cell factor (SCF) has been demonstrated to optimize the proliferation and subsequent differentiation of progenitors into adherent macrophages under serum-free conditions [63].

Final Maturation (Approximately After Day 21): Mature macrophages express characteristic surface markers such as CD11b, CD14, CD64, CD115, or CD163 and exhibit typical macrophage functions including phagocytosis and inflammatory responses [59,60,62,63]. The macrophages generated closely resemble in vivo macrophages, making them suitable for research applications including disease modeling and immune studies.

Protocol Variations and Considerations: The OP9 stromal cell co-culture system has several significant advantages. The stromal cells provide niche signals enhancing hematopoietic differentiation efficiency, allowing the generation of large numbers of functional macrophages. However, there are notable limitations. Variability between stromal cell batches can lead to issues in reproducibility, posing challenges for consistent outcomes. Furthermore, using mouse-derived OP9 cells introduces undefined xenogeneic factors, potentially limiting their clinical application. The use of FBS instead of defined factors is also a relative disadvantage. The dependence on stromal feeder layers complicates standardization relative to feeder-free methods. 

#### 4.1.3. Embryoid Body (EB) Formation

The embryoid body (EB) method is a widely used approach to differentiate hPSCs, encompassing both hESCs and iPSCs, into macrophages. This protocol mirrors early yolk sac hematopoiesis through a stepwise progression: mesoderm induction, hematopoietic specification, progenitor expansion, and macrophage maturation (Figure 5) [64,65,66,67,68,69,70,71,72].

Embryoid Body (EB) Formation and Mesodermal Induction (Approximately Day 0–4): The differentiation starts with inducing hPSCs into the mesodermal lineage. hPSCs are cultured in EB differentiation medium, typically containing bone morphogenetic protein 4 (BMP4), vascular endothelial growth factor (VEGF), and stem cell factor (SCF) [66,67,69,73]. BMP4 promotes the formation of mesoderm. VEGF supports endothelial and hematopoietic commitment. SCF maintains survival and proliferation of mesodermal precursors. Notably, Atkins et al. highlight the critical role of fibroblast growth factor 2 (FGF2) alongside Activin A during early mesoderm specification, enhancing hematopoietic commitment and closely resembling yolk sac development. The hPSCs are plated on ultra-low attachment plates, allowing spontaneous EB formation. Using feeder-free and defined conditions helps maintain reproducibility and consistency.

Hematopoietic Specification (Approximately Day 4–8): Following mesoderm induction, EBs are transferred onto adherent surfaces to guide differentiation toward hematopoietic lineage [66,67,69,73]. The culture medium at this stage commonly includes VEGF, SCF, and FMS-like tyrosine kinase 3 ligand (FLT3L). VEGF supports differentiation into hemogenic endothelium (HE). SCF and FLT3L assist in the emergence and expansion of early hematopoietic progenitors. This step mimics the embryonic transition from mesoderm to hemogenic endothelium, the essential precursor to blood and immune cells.

Hematopoietic Progenitor Expansion (Approximately Day 8-15): Emerging hematopoietic progenitors are then expanded to increase their numbers. During this phase, cells are cultured in suspension condition using hematopoietic expansion medium. Commonly included cytokines at this stage are SCF, interleukin-3 (IL-3), macrophage colony-stimulating factor (M-CSF), and FLT3L [66,67,69]. SCF and FLT3L promote progenitor survival. IL-3 supports progenitor proliferation, and M-CSF directs cells toward the myeloid lineage. Suspension culture conditions help sustain progenitor cell survival and expansion, preparing them for further differentiation into macrophages. In this stage, erythro-myeloid progenitor-like cells expressing markers such as CD41 and CD45, paralleling the primitive yolk sac hematopoietic progenitors [72].

Macrophage Differentiation (Approximately Day 15–22): At this stage, cells are driven toward macrophage differentiation. The progenitors are transferred to adherent culture conditions using macrophage differentiation medium supplemented with M-CSF. Optionally, interleukin-34 (IL-34) can also be added. M-CSF is essential for macrophage lineage commitment, while IL-34 promotes the development of tissue-resident macrophage characteristics. At this stage, feeder layers, if previously used, are depleted to achieve uniform differentiation [64,66,67].

Final Maturation (Approximately After Day 22): In the final stage, macrophages mature fully into functional cells. Macrophage maturation medium contains M-CSF, IL-34, and tissue-specific cytokines, depending on the desired macrophage subtype. Cells remain adherent, acquiring functional phenotypes. Mature macrophages are characterized by the expression of established macrophage markers (CD11b, CD14, CD68, CD163), phagocytic activity, and responsiveness to inflammatory cues, resembling tissue-resident macrophages [64,65,66,67,68,69,70,71].

Protocol Variations and Considerations: Advantages of EB-based differentiation include its resemblance to early embryonic hematopoiesis. EB-derived macrophages closely resemble yolk sac-derived macrophages, such as microglia, Kupffer cells, and alveolar macrophages [67,72]. Another advantage is scalability, as many EB protocols allow sustained macrophage production for extended periods. This makes the EB method a valuable and renewable resource for long-term studies [65,66,69]. Additionally, this differentiation method facilitates genetic modifications. Genetic changes can be introduced at the hPSC stage, allowing analysis of gene functions in differentiated macrophages, valuable for disease modeling.

Despite these advantages, the EB-based method has some limitations. One significant challenge is variability in differentiation efficiency due to inconsistent EB size and composition, affecting macrophage yield and purity [66,69]. Another concern is the presence of undefined components, such as serum or feeder layers in some protocols, leading to batch-to-batch variability. Nevertheless, the EB-based differentiation method remains effective and popular. Ongoing research continues to refine the process, focusing on serum-free and feeder-free conditions, aiming for improved scalability, purity, and clinical applicability.

### 4.2. Polarization of hPSC-Derived Macrophages (hPSC-Ms)

Human pluripotent stem cell-derived macrophages (hPSC-Ms) can be polarized into distinct functional states, commonly classified as M1 and M2 macrophages, depending on the specific environmental stimuli they receive [49,68,74]. The M1 macrophage phenotype, known for its pro-inflammatory properties, is commonly induced by treating hPSC-Ms with interferon-gamma (IFN-γ) combined with lipopolysaccharide (LPS) [49,68,74]. These stimuli activate macrophages to produce cytokines such as tumor necrosis factor-alpha (TNF-α), interleukin-6 (IL-6), and nitric oxide (NO). M1 macrophages significantly enhance antimicrobial activities and pathogen elimination by increasing inflammatory cytokines and nitric oxide production. They also demonstrate elevated phagocytic capacities directed specifically toward pathogens, aiding their clearance.

Conversely, the anti-inflammatory M2 macrophage phenotype (specifically the M2a subtype) is typically induced by interleukin-4 (IL-4) and interleukin-13 (IL-13), often in the presence of macrophage colony-stimulating factor (M-CSF) [49,68,74]. These cytokines promote the upregulation of specific markers associated with M2 polarization, particularly the mannose receptor (CD206). M2 macrophages actively participate in immune regulation and tissue repair. They effectively reduce inflammation and promote fibrosis, wound healing, and tissue remodeling.

### 4.3. Challenges and Future Directions

Generating macrophages from hPSCs follows a stepwise process. Each method of approach—monolayer 2D cell culture, co-culture with stromal cells, and embryoid body (EB) formation—has strengths and weaknesses (Table 1). 

Standardization remains a challenge. Many differentiation protocols use different cytokine combinations, timing, and culture conditions. Variability in hPSC, such as using iPSC versus hESC, can affect differentiation outcomes and macrophage function. For example, some protocols generate macrophages within 7–14 days, while others extend maturation beyond 21 days. Differences in culture media, feeder layers, and small molecules further contribute to batch-to-batch variability.

Transcriptomic differences between hPSC-Ms and monocyte-derived macrophages (MDMs) present another challenge [75,76,77]. hPSC-Ms express more extracellular matrix (ECM)-related genes and fewer major histocompatibility complex class II (MHC-II) genes. These differences impact immune response and antigen presentation. Future studies should focus on optimizing culture conditions to better replicate primary macrophage phenotypes.

Functional maturity is another concern. Many hPSC-Ms share key markers with tissue-resident macrophages, but they may not fully replicate tissue-specific functions. Additional cues, such as co-culture with organ-specific cells, has the potential enhance functional specialization [78,79,80,81,82].

Scalability is another important consideration. Some EB-based protocols allow continuous macrophage production for several months. This feature makes hPSC-Ms attractive for disease modeling and drug screening. However, maintaining long-term cultures requires optimization to ensure reproducibility.

Clinical applications remain limited. Most protocols still rely on undefined components, such as fetal bovine serum (FBS) or feeder cells. Eliminating these elements will improve consistency and regulatory approval for therapeutic use. Advances in serum-free and feeder-free culture systems should make hPSC-Ms more suitable for clinical applications.

## 5. Methods to Generate hECTs

Human engineered cardiac tissues (hECTs) are three-dimensional constructs that closely mimic native heart tissue. These tissues are generated by embedding human pluripotent stem cell-derived cardiomyocytes (hPSC-CMs) into various supportive environments. The key methods include scaffold-based approaches, physical conditioning, miniaturized scalable models, chamber-specific models, immune cell incorporation, and disease modeling [12,83,84,85,86,87,88,89,90,91,92,93,94,95,96,97].

### 5.1. Scaffold-Based and Biohybrid Approaches

Natural hydrogels commonly provide structural support in cardiac tissue engineering. Hydrogels such as fibrin, collagen, and decellularized cardiac extracellular matrix (ECM) allow cells to organize naturally into functioning cardiac tissue [12,13,85,87,89,96,97,98]. 

Fibrin is the most commonly used hydrogel in hECT models. It supports cell viability, promotes tissue compaction, and enables spontaneous and paced contractions. Most platforms embed hPSC-CMs and cardiac fibroblasts in fibrin. These cell-laden hydrogels are cast into molds to form beating tissues. Collagen is also widely used. It offers strong mechanical support and promotes alignment of cardiomyocytes. In some systems, collagen is combined with fibrin or enhanced with decellularized cardiac extracellular matrix (ECM). These modifications improve the biochemical environment and support structural maturation. For example, one model embeds atrial or ventricular hPSC-CMs in collagen to form chamber-specific rings. These tissues retain distinct electrical activity and contraction patterns, depending on the cell type used [85].

### 5.2. Electrical and Mechanical Conditioning Models

Physical stimulation significantly enhances the maturity and function of cardiac tissues. Electrical and mechanical conditioning methods have become standard techniques. One widely used platform is the Biowire system [12,89,93,97,98]. In this method, hPSC-CMs and supporting cells are embedded in a fibrin or collagen gel. The gel is cast between two flexible wires, and electrical pacing is applied. This stimulation improves sarcomere organization, increases calcium handling, and enhances conduction velocity.

Intensity training regimens have also been developed to promote maturation. This protocol gradually increases pacing frequency from 2 Hz to 6 Hz over several weeks [13,87]. Tissues trained in this way develop more mature properties such as a force frequency relationship. They display organized sarcomeres, dense mitochondria, and mature electrical behavior. They also respond to drugs in a way that closely mimics adult human heart tissue.

Mechanical conditioning can also support tissue development. The I-Wire platform suspends the tissue between two posts, allowing controlled stretch and measurement of force. This setup helps characterize both passive and active mechanical properties of the tissue [94].

### 5.3. Miniaturized and Scalable Models

High-throughput drug screening and disease modeling require miniaturized cardiac tissues. These platforms use fewer cells per tissue and allow the facile and simultaneous testing of many conditions at once. The micro-heart muscle (µHM) platform uses about 2000 hPSC-CMs per construct [90]. These cells are embedded in fibrin and cast into small molds. Despite their size, the tissues show strong contraction and reproducible drug responses. They also demonstrate a force–length relationship and respond to β-adrenergic stimulation.

The Biowire II platform automates tissue fabrication. It uses 3D-printed microwires and carbon electrodes to pace tissues in parallel. The system supports 8-, 24-, or 96-well formats. Tissues made in this platform show positive force–frequency relationships and are suitable for long-term pacing and drug screening [89].

Another approach uses bio-3D printing to create scaffold-free cardiac tubes [83]. Spheroids made from hPSC-CMs and fibroblasts are printed into tubular constructs. These tissues fuse over time and develop coordinated contractions. They offer a scalable alternative to hydrogel-based methods.

### 5.4. Disease Modeling with hPSC-CMs and hECTs

Human engineered cardiac tissues (hECTs) provide a powerful platform for modeling both inherited and acquired cardiac diseases in a human-specific, physiologically relevant environment. These models enable mechanistic studies, therapeutic screening, and precision medicine approaches by capturing patient-specific phenotypes in vitro.

The most successful applications of these systems have been in genetically driven cardiomyopathies. Multiple functional readouts from induced pluripotent stem cell–derived cardiomyocytes (hPSC-CMs), in both two-dimensional (2D) and three-dimensional (3D) formats, show strong correlations with clinical phenotypes. These models also facilitate therapeutic screening using low- to medium-throughput platforms for assessing cardiomyocyte responses.

Dilated cardiomyopathy (DCM) was first modeled using hPSC-CMs carrying a troponin T type 2 (TNNT2) R173W mutation in monolayer culture, with contractile properties quantified by atomic force microscopy [99]. The DCM phenotype—characterized by reduced contractile force, disorganized sarcomeres, and impaired calcium handling—was evident even at the single-cell level.

Hypertrophic cardiomyopathy (HCM) has been modeled using both 2D and 3D systems. An early study employed 2D monolayers of hPSC-CMs carrying a myosin heavy chain 7 (MYH7) R663H mutation, identifying calcium cycling abnormalities and cellular hypertrophy [100]. More recently, researchers developed a 3D HCM model using hECTs derived from patient-specific hPSC-CMs with an MYH7 R403Q mutation [101]. These tissues, generated using the Biowire platform to promote electrical maturation, displayed increased size, sarcomere disorganization, prolonged action potentials, and abnormal calcium handling. Treatment with the myosin inhibitor mavacamten reversed several disease features, including impaired relaxation and elevated B-type natriuretic peptide (BNP) expression, demonstrating the therapeutic responsiveness of this system.

Restrictive cardiomyopathy (RCM) was modeled using 3D fibrin-based hECTs generated from hPSC-CMs carrying a pathogenic filamin C (FLNC) mutation [102]. These tissues exhibited impaired relaxation velocity and increased passive stiffness—key clinical features of RCM. The model further identified phosphodiesterase 3 inhibition as a potential therapeutic approach for FLNC-associated RCM.

Inherited arrhythmias have also been successfully modeled. A seminal study on LQT type 1 (LQT1) employed hPSC-CMs carrying a potassium voltage-gated channel subfamily Q member 1 (KCNQ1) R190Q mutation in 2D monolayers [103]. These cells showed prolonged action potentials, reduced slow delayed rectifier potassium current (IKs), and exaggerated adrenergic responses.

For LQT type 2 (LQT2), modeling has been conducted in both 2D and 3D systems [104,105,106]. Early studies used hPSC-CMs with a KCNH2 R176W mutation in monolayer culture, revealing prolonged repolarization, reduced rapid delayed rectifier potassium current (IKr), and heightened sensitivity to arrhythmogenic agents such as sotalol. More recent work introduced a KCNH2 R531W mutation via CRISPR-Cas9 into isogenic hPSC-CMs, which were co-cultured with human cardiac fibroblasts in a collagen hydrogel to form anisotropic 3D hECTs within a heart-on-a-chip microfluidic platform [105]. These engineered tissues recapitulated key LQT2 features, including defective hERG channel trafficking, reduced IKr, abnormal calcium transients, and increased beat interval variability under catecholaminergic stimulation.

LQT type 3 (LQT3) was modeled using hPSC-CMs carrying an SCN5A R1644H mutation, cultured as monolayers [107]. These cells demonstrated early afterdepolarizations, prolonged sodium current, and increased arrhythmia susceptibility, which were reversed with sodium channel blockers.

Timothy syndrome (LQT8), caused by mutations in calcium voltage-gated channel subunit alpha 1C (CACNA1C), was modeled in embryoid body–derived monolayer hPSC-CMs. These cells showed prolonged action potentials, irregular calcium transients, and spontaneous arrhythmias [108].

Barth syndrome, a rare X linked metabolic disorder that leads to skeletal muscle weakness as well as dilated cardiomyopathy has also been modeled using hPSC-CMs carrying mutations in tafazzin (TAZ) [92]. These tissues exhibited impaired contractility, sarcomere disorganization, and increased reactive oxygen species. Notably, restoring TAZ expression or reducing oxidative stress rescued these phenotypes.

hECTs have further been applied to model acquired heart diseases. For example, myocardial infarction (MI) has been simulated by exposing hECTs to hypoxia and chronic noradrenaline stimulation, creating spatial gradients of injury across the tissue that mimic infarct, border, and remote zones in the human heart [86,88]. These infarct organoids exhibit hallmark features of MI—including fibrosis, metabolic shifts, and calcium dysregulation—with transcriptomic profiles closely matching those of human ischemic tissue, making them valuable tools for drug screening and cardiotoxicity testing. Additionally, the autoimmune cardiac damage observed in systemic lupus erythematosus (SLE) was modeled by applying autoantibodies derived from patient sera directly to engineered tissues [87]. Under stress, these antibodies bound to apoptotic blebs or cardiomyocytes themselves, impairing calcium handling, altering gene expression, and reducing tissue contractility. This work demonstrated how circulating antibodies can directly induce myocardial injury independent of immune cells.

### 5.5. Integration of Artificial Intelligence for Functional Phenotyping

Artificial intelligence (AI) is becoming an important tool in cardiac tissue engineering. It helps researchers analyze complex high dimensional data from hECTs more effectively. One example is BeatProfiler, a machine learning framework that classifies cardiac tissue phenotypes based on multiple types of functional data. Kim et al. developed this multimodal platform that analyzes both contractile motion and calcium transient data from hECTs [109]. These tissues were derived from hPSCs and modeled cardiac dysfunction caused by a FLNC mutation or drug treatments. BeatProfiler captures videos of tissue contraction and calcium dynamics in a synchronized format, allowing high-throughput, label-free assessment of cardiac function. Compared to existing tools, it reduces false positives and performs better on low-signal data. The authors used two machine learning approaches to classify disease and drug conditions. A feature-based model using principal component analysis with k-nearest neighbors achieved 93.3% accuracy. A deep learning model using a temporal convolutional network with bidirectional long short-term memory reached 96.5% accuracy. Both models could distinguish between healthy and disease tissues and differentiate the effects of verapamil, isoproterenol, and propranolol. This study shows that AI can extract subtle phenotypic patterns from complex datasets that may not be obvious to human observers. By combining hECTs with AI-driven functional profiling, researchers can now classify cardiac disease states and drug effects with greater speed and accuracy.

## 6. Inclusion of Resident Macrophages in hPSC-Derived hECTs

hECTs are important models for studying heart function and disease. Traditionally, these tissues mainly consist of hPSC-CM. However, incorporating only cardiomyocytes limits the ability to fully mimic the complex structure and function of native heart tissue. To address this limitation, recent studies have explored macrophage inclusion in hECTs. Three recent studies addressed this limitation by including macrophages in hECTs (Table 2) [97,98,110]. Lock et al. (2024), Landau et al. (2024), and Hamidzada et al. (2024) explored different ways to incorporate macrophages into engineered cardiac models [97,98,110]. 

Lock et al. (2024) created hECTs by differentiating iPSCs into cardiomyocytes, fibroblasts, and macrophages [110]. They produced cardiomyocytes by modulating the Wnt signaling pathway. Cardiomyocytes were validated using markers like cardiac troponin T (cTnT) and alpha-actinin. Fibroblasts were identified by markers such as vimentin, transcription factor 21 (TCF21), and GATA binding protein 4 (GATA4). Macrophages were generated by first differentiating iPSCs into hematopoietic progenitor cells, which subsequently matured into macrophages. These macrophages were confirmed using markers like cluster of differentiation 68 (CD68) and cluster of differentiation 11b (CD11b). These cells were embedded in a fibrin hydrogel matrix and cultured for approximately 28 days. Increased contractile force was reported with macrophage incorporation. Engineered tissues with macrophages produced twice as much force as those without. This enhanced contractility resulted primarily from macrophages activating the beta-adrenergic signaling pathway in cardiomyocytes. Specifically, they observed increased expression of beta-adrenergic receptor 1 (ADRB1) and adenylyl cyclase 3 (ADCY3). Additionally, macrophage-containing tissues demonstrated improved calcium handling, enabling faster contraction and relaxation. These tissues also secreted higher levels of inflammatory cytokines like interleukin-6 (IL-6) and tumor necrosis factor-alpha (TNF-alpha), as well as pro-regenerative cytokines like interleukin-10 (IL-10).

Landau et al. (2024) used human embryonic stem cells (hESCs) to generate cardiac tissues [98]. hESCs were differentiated into primitive macrophages, resembling those from the embryonic yolk sac, using the EB formation method. These primitive macrophages were distinct from macrophages derived from hematopoietic lineages. These macrophages were combined with endothelial cells, stromal cells, and cardiomyocytes in engineered cardiac tissues. This model emphasized creating stable, vascularized cardiac structures. In these vascularized heart-on-a-chip models, macrophages promoted the formation of stable, functional blood vessels. RNA sequencing revealed that macrophages stimulated stromal cells to release pro-angiogenic factors, including insulin-like growth factor binding protein 7 (IGFBP7) and hepatocyte growth factor (HGF). Perfusion experiments confirmed that tissues with macrophages maintained functional microvascular networks for extended periods. This suggests that macrophages play a key role in supporting blood vessel health in engineered tissues.

Hamidzada et al. (2024) developed cardiac microtissues using hESCs [97]. Primitive macrophages were generated through protocols mimicking early embryonic development, closely resembling macrophages derived from the embryonic yolk sac using the EB formation method. These macrophages were characterized using markers such as lymphatic vessel endothelial hyaluronan receptor 1 (LYVE1), cluster of differentiation 14 (CD14), CD68, and CD163. Cells were combined within a hydrogel matrix in a heart-on-a-chip platform called Biowire. These macrophages integrated stably within the cardiac microtissues. The incorporation of macrophages resulted in significant functional enhancements, including improved contractile force and relaxation kinetics. The macrophages facilitated cardiomyocyte maturation by promoting sarcomeric protein development, specifically troponin and tropomyosin complexes, as well as myosin heavy chain proteins. Mechanistically, macrophages performed efferocytosis, clearing apoptotic cardiomyocyte debris through phosphatidylserine-dependent mechanisms. This process reduced tissue stress and enhanced cardiomyocyte maturation, aligning their transcriptional and metabolic profiles closely with early human fetal ventricular cardiomyocytes. Blocking macrophage efferocytosis impaired these beneficial effects, confirming the critical role of macrophages in cardiac microtissue function and maturation.

## 7. Future Directions

Classic therapeutic development has relied on animal models which has yielded of human efficacy. Engineered cardiac tissue holds great promise in providing a physiologically relevant and human-based model for heart disease to add improved specificity for target identification and drug development. This review highlights the potential for incorporating immune cells in engineered cardiac models. In particular, macrophages play an important role in cardiac injury and the incorporation of these cells into ECTs has begun with some early success. This is an important step towards replicating the endogenous cellular diversity of the heart yet there remains other opportunities within the immune system that can be leveraged in the future using hPSC technology. Specifically, regulatory T cells (Tregs) and dendritic cells (DCs) are now derivable from iPSCs and play key roles in heart disease. 

iPSC-Tregs hold promise for enhancing hECTs by modulating inflammation and fibrosis. Tregs are a specialized subset of CD4⁺ T cells that suppress excessive immune responses and help maintain immune tolerance. These cells are known for promoting tissue repair and dampening inflammation. Tregs in the heart can reduce harmful post-infarct inflammation and have been shown to decrease cardiac fibrosis [111,112]. For instance, studies indicate that Tregs secrete anti-inflammatory cytokines like IL-10, which can directly attenuate fibrotic pathways and influence fibroblast behavior. The incorporation of iPSC-Tregs into hECTs would mimic the heart’s natural “brake” on inflammation, potentially yielding models that develop more physiologic scar tissue and stable healing. This inclusion would be especially valuable in models of myocardial infarction or chronic heart failure, where Tregs are believed to help form a stable scar early but later become dysfunctional and pro-fibrotic in chronic disease. iPSC technology makes it feasible to derive Tregs in vitro by inducing FOXP3 expression in iPSC-derived CD4⁺ cells, generating stable populations of suppressive Tregs [113,114]. These iPSC-Tregs can be added to cardiac microtissues to study how they affect cardiomyocyte survival, electrophysiology, and fibrosis over time. 

Dendritic cells are pivotal antigen-presenting cells that orchestrate immune responses and maintain tissue homeostasis. In engineered cardiac tissues, iPSC-DCs could serve as sentinels that facilitate communication between innate and adaptive immune cells. For example, mature DCs present antigens via MHC molecules and provide co-stimulation to T cells, which could be harnessed in hECTs to study autoimmune myocarditis or responses to tissue injury [115]. Including iPSC-DCs in hECTs may allow us to model how the heart’s resident antigen-presenting cells educate T cells in both healthy and diseased states. Supporting studies in immuno-engineered organoids have shown that adding DCs enhances immune cell recruitment and functional responses [116]. In cardiac models, iPSC-DCs may help recruit Tregs or other T cell subsets into the tissue to release cytokines influencing macrophage polarization and cardiomyocyte behavior. Another key role of DCs in the heart involves inducing tolerance and resolving inflammation; a subset of “tolerogenic” DCs can drive Treg differentiation via IL-10 secretion [117]. Integrating iPSC-DCs with hECTs would aid studies of this tolerogenic axis under controlled conditions. 

A promising future direction is the use of hECTs to model autoimmune-related cardiac injury. A recent study demonstrates that systemic lupus erythematosus (SLE) autoantibodies can bind to cardiomyocytes, altering calcium handling, respiration, and mitochondrial function, ultimately leading to ventricular dysfunction and heart failure [87]. Incorporating a variety of immune cells will allow researchers to explore how autoantibodies trigger inflammation, fibrosis, and cardiomyocyte dysfunction. Additionally, patient-derived hECTs could be used to study individual variations in autoimmune cardiac disease by profiling autoantibody targets and their effects on cardiac cells. Future studies should investigate how macrophages influence autoantibody responses in hECTs and whether specific Treg subsets can suppress autoimmune-driven fibrosis.

Despite these potential advances, technical challenges remain. Current engineered cardiac tissue models still struggle with reproducibility in terms of cellular composition and structural organization [118]. Ensuring that hPSC-derived immune cells integrate properly within the three-dimensional cardiac microenvironment remains difficult, as differences in cytokine gradients, cell–cell interactions, and tissue maturation affect their functionality. Additionally, matured pathogenic cardiac constructs continue to pose a significant challenge [119]. Aging profoundly alters immune cell function, and accurately modeling an aged or diseased myocardium in vitro requires fine-tuned culture conditions that recapitulate age-associated changes in immune signaling, extracellular matrix composition, and cardiomyocyte function [120].

A major question for future research is whether these advanced platforms can faithfully replicate immune-cardiac cross-talk under both physiological and pathological conditions. If immune cell infiltration and disease-specific phenotypes can be accurately engineered, these models will become powerful tools for studying myocardial infarction, heart failure, arrhythmias, and other immune-mediated cardiac diseases. Additionally, such platforms will facilitate the development of patient-specific disease models, allowing for personalized testing of immunotherapies and regenerative treatments. There is also the potential to develop regenerative therapies, including cardiac grafts preloaded with immune cells. Such grafts could reduce rejection and improve functional integration after transplantation.

Ultimately, incorporating multiple hPSC-derived immune cell types into engineered cardiac tissues will push the boundaries of cardiovascular research. These models will bridge the gap between traditional in vitro studies and in vivo complexities, providing a more scalable yet accurate system to study heart disease mechanisms and potential therapies. While challenges remain in tissue reproducibility and immune integration, ongoing advancements in stem cell differentiation, tissue engineering, and immunology will continue to refine these platforms, bringing us closer to fully functional, immune-competent cardiac constructs for research and therapeutic applications.

## 8. Conclusions

The hECTs are changing how scientists study heart disease in the laboratory. This review focused on how including macrophages, especially those derived from hPSCs, improves the accuracy of these models. Macrophages help clear apoptotic debris, regulate inflammation, and support tissue remodeling. They are essential in capturing immune-cardiac interactions that occur after myocardial injury. New scaffold designs, mechanical and electrical conditioning, and co-cultures with immune cells have brought hECTs closer to mimicking the human heart.

Despite these advances, challenges remain. It remains challenging to ensure the consistent integration of macrophages into three-dimensional tissues. Current models also lack age-related or disease-specific immune characteristics. Including other immune cells such as Tregs and DCs may further improve the physiological relevance of hECTs. These additions can help model chronic inflammation, fibrosis, and autoimmunity more accurately.

Moving forward, combining multiple immune cell types with hPSCs will allow researchers to build more complex and human-relevant cardiac models. These platforms can provide insights into heart failure, arrhythmias, and autoimmune heart disease. They may also help in developing personalized therapies. As stem cell differentiation and tissue engineering continue to improve, immune-competent hECTs will become powerful tools in cardiovascular research and medicine.

## Figures and Tables

**Figure 1 cells-14-01393-f001:**
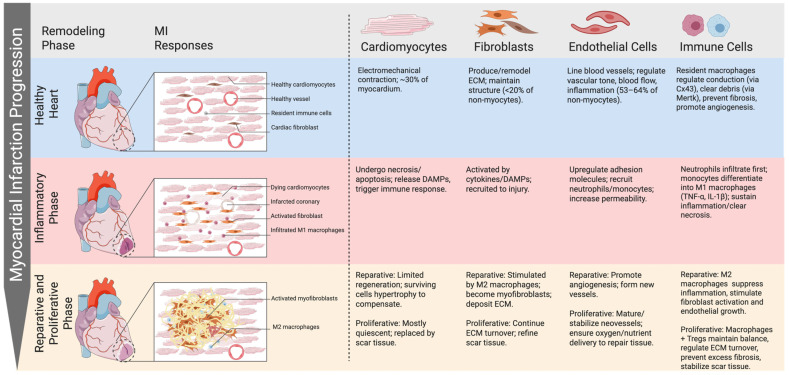
**Phases of myocardial infarction and the role of each cell type.** Cardiac remodeling after myocardial infarction progresses through inflammatory, reparative, and proliferative phases. Cardiomyocytes undergo necrosis and release DAMPs, activating immune responses. Neutrophils and M1 macrophages dominate the early inflammatory stage, while fibroblasts and endothelial cells contribute to scar formation and angiogenesis. During the reparative/proliferative phases, M2 macrophages and Tregs suppress inflammation, promote ECM deposition, and stabilize neovessels, ensuring balanced healing and scar maturation.

**Figure 2 cells-14-01393-f002:**
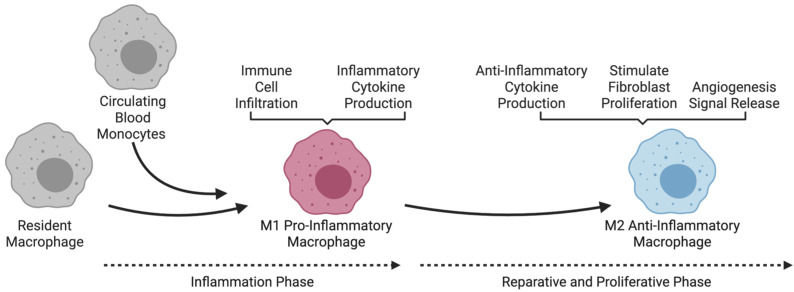
**Function of macrophages during myocardial infarction**. During myocardial infarction, resident macrophages and circulating monocytes first polarize into M1 macrophages, producing inflammatory cytokines in the early inflammatory phase. As repair progresses, macrophages transition to the M2 phenotype, which resolves inflammation, stimulates fibroblast activity, and promotes angiogenesis during the reparative and proliferative phases.

**Figure 3 cells-14-01393-f003:**
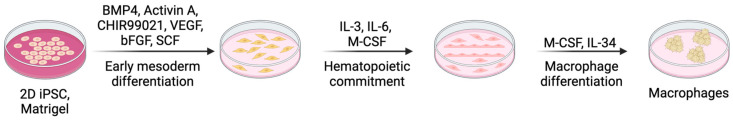
**iPSC-derived macrophages: monolayer 2D cell culture**. Schematic representation of the generation of macrophages from pluripotent stem cells using monolayer 2D cell culture method. iPSCs are cultured on Matrigel in feeder-free conditions to generate macrophages. The process includes mesoderm induction (BMP4, Activin A, CHIR99021), hematopoietic specification (VEGF, bFGF, SCF), progenitor expansion (VEGF, SCF, IL-3, IL-6), and macrophage differentiation and maturation (M-CSF, IL-34). Resulting cells express CD11b, IBA1, CD14, CD68, TMEM119, and P2RY12, resembling primary human macrophages.

**Figure 4 cells-14-01393-f004:**
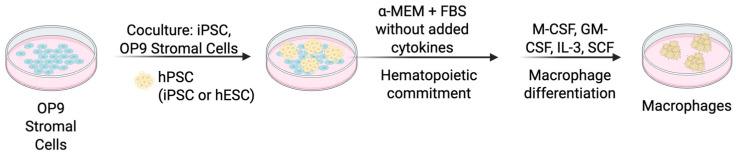
**hPSC-derived macrophages: co-culture with stromal cells**. Schematic representation of the generation of macrophages from pluripotent stem cells using co-culture with stromal cells method. hPSCs are seeded onto a confluent OP9 stromal layer to induce mesoderm and hematopoietic specification (Day 0–9). Hematopoietic progenitors are harvested and cultured with M-CSF (±GM-CSF) to promote macrophage lineage commitment (after Day 12), followed by final maturation (after Day 21).

**Figure 5 cells-14-01393-f005:**
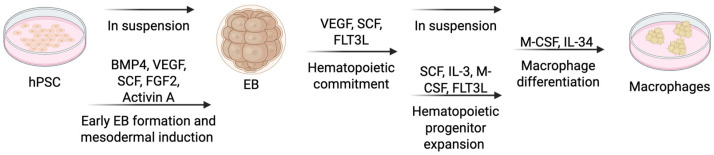
**hPSC-derived macrophages: co-culture with stromal cells**. Schematic representation of the generation of macrophages from pluripotent stem cells using embryoid body (EB) formation method. Stepwise differentiation of macrophages from human pluripotent stem cells (hPSCs) via embryoid body (EB) formation. The process mimics yolk sac hematopoiesis and includes mesoderm induction (BMP4, VEGF, SCF, FGF2), hematopoietic specification (VEGF, SCF, FLT3L), progenitor expansion (IL-3, M-CSF, SCF, FLT3L), macrophage differentiation (M-CSF, IL-34), and terminal maturation. Mature macrophages express canonical markers (CD11b, CD14, CD68, CD163), demonstrate phagocytic activity, and share MYB-independent ontogeny with yolk-sac-derived tissue-resident macrophages.

**Table 1 cells-14-01393-t001:** Comparison of hPSC-Ms differentiation methods.

Differentiation Method	Key Characteristics	Advantages	Limitations
Monolayer (2D) Culture	Stepwise differentiation using defined media; cytokines (BMP4, VEGF, SCF, IL-3, IL-6, M-CSF, IL-34)	Precise control, reproducible, efficient	Requires optimization, variability among iPSC lines, limited maturation
Co-Culture with Stromal Cells (OP9)	Stromal cell interactions drive differentiation; minimal cytokine use initially	High yield, functional macrophages, mimics niche signaling	Batch variability, xenogeneic concerns (mouse cells), undefined factors
Embryoid Body (EB) Formation	Mimics yolk sac hematopoiesis; cytokines (BMP4, VEGF, SCF, FGF2, FLT3L, IL-3, M-CSF, IL-34) used	Resembles embryonic development, scalable, facilitates genetic editing	Variable EB size, inconsistent yields, undefined components (e.g., serum)

**Table 2 cells-14-01393-t002:** Inclusion of macrophages in hECTs.

Study	Cell Types Included	Key Findings	Implications
Lock et al. (2024)	iPSC-derived cardiomyocytes, fibroblasts, macrophages (hematopoietic origin)	Enhanced contractile force; improved calcium handling and beta-adrenergic signaling	Physiologically relevant cardiac models; useful in regenerative medicine and drug testing
Landau et al. (2024)	Human embryonic stem cell (hESC)-derived cardiomyocytes, endothelial cells, stromal cells, primitive macrophages (by EB Formation; yolk sac-like origin)	Improved vascularization and vessel stability; increased pro-angiogenic signaling	Effective modeling of cardiac vascular networks; potential for improving tissue integration post-transplantation
Hamidzada et al. (2024)	hESC-derived cardiomyocytes, fibroblasts, primitive macrophages (by EB Formation; yolk sac-like origin)	Enhanced contractility and relaxation; promoted sarcomeric maturation via efferocytosis	Improved cardiac tissue maturation; valuable for modeling developmental processes and regenerative therapies

References: [97,98,110].

## Data Availability

No new data were created or analyzed in this study.

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
