# Peer review of "Integrating Macrophages into Human-Engineered Cardiac Tissue"

_cells, 2025, doi:10.3390/cells14171393_

Round 1
Reviewer 1 Report
Comments and Suggestions for Authors
In this article the authors assess the topic Integrating Macrophages into Human-Engineered Cardiac Tissue
This is an intersting topic and my minor concerns are the following
- what is the clinical implication of the findings?
- What is the role of genetics?
- What is the role of AI?
- What about ischemic cardiomyopathy?
- What is the role for the preserved Heart failure
- What is the role in myocarditis?
Reviewer 2 Report
Comments and Suggestions for Authors
Review is focused on an actual and important topic. I have the following comments:
1. The Figures 2 and 3 do not representatively capture the individual phases described in the corresponding text.
Moreover, the legend to these Figures with more information is missing.
2. In lines 139-141 you wrote: “Studies have demonstrated that depletion of resident macrophages leads to increased fibrosis and impaired angiogenesis, underscoring their protective role during cardiac stress.”
The references for these studies are not presented.
3. In lines 468-471 you wrote: “More recent work introduced a KCNH2 R531W mutation via CRISPR-Cas9 into isogenic hPSC-CMs, which were co-cultured with human cardiac fibroblasts in a collagen hydrogel to form anisotropic 3D hECTs within a heart-on-a-chip microfluidic platform.”
The reference for this study is not presented.
Reviewer 3 Report
Comments and Suggestions for Authors
The review "Integrating Macrophages into Human-Engineered Cardiac Tissue" summarizes the recent developments and findings of human-based engineered cardiac tissues (hECTs) using novel scaffolds and pluripotent stem cell derivatives, along with the advantages of co-culturing them with macrophages. The review is comprehensive and well written. I have the following suggestions.
- The abstract mentions that "This review explores new methods for deriving macrophages 16 from human pluripotent stem cells," however, this is not the case. This review summarizes the findings.
- Please cite lines 52-53.
- These phases 66 involve highlight the critical role immune cells play- please check for sentence formation.
- Please include a figure for the three phases of cardiac repair.
- Cardiac resident macrophages have been discussed; what about infiltrated macrophages?
- Please cite the findings in Table 1.
- Please cite the lines 344-248.
- Please cite the three studies in Table 2.
- Please include a conclusion section.
- Please include more figures, mainly in section 3, and one figure explaining the function and how these macrophages, when co-cultured, will be beneficial.
